# Thermal management and non-reciprocal control of phonon flow via optomechanics

Alireza Seif[1,2], Wade DeGottardi[1,2,3], Keivan Esfarjani[4,5,6] & Mohammad Hafezi [1,2,3]

Engineering phonon transport in physical systems is a subject of interest in the study of materials, and has a crucial role in controlling energy and heat transfer. Of particular interest are non-reciprocal phononic systems, which in direct analogy to electric diodes, provide a directional flow of energy. Here, we propose an engineered nanostructured material, in which tunable non-reciprocal phonon transport is achieved through optomechanical coupling. Our scheme relies on breaking time-reversal symmetry by a spatially varying laser drive, which manipulates low-energy acoustic phonons. Furthermore, we take advantage of developments in the manipulation of high-energy phonons through controlled scattering mechanisms, such as using alloys and introducing disorder. These combined approaches allow us to design an acoustic isolator and a thermal diode. Our proposed device will have potential impact in phonon-based information processing, and heat management in low temperatures.

[1] Joint Quantum Institute, NIST/University of Maryland, College Park, MD 20742, USA. [2] Department of Physics, University of Maryland, College Park, MD 20742, USA. [3] Department of Electrical and Computer Engineering and Institute for Research in Electronics and Applied Physics, University of Maryland, College Park, MD 20742, USA. [4] Department of Mechanical and Aerospace Engineering, University of Virginia, Charlottesville, VA 22904, USA. [5] Departments of Materials Science and Engineering, University of Virginia, Charlottesville, VA 22904, USA. [6] Department of Physics, University of Virginia, Charlottesville, VA 22904, USA. Correspondence and requests for materials should be addressed to M.H. (email: hafezi@umd.edu)

Controlling the flow of heat is important for several fields including thermoelectrics, thermal management, and information processing. For example, suppressing thermal conductivity can improve the performance of thermoelectrics, and can also isolate circuit elements from external heat. The thermal analog of an electric diode is of fundamental importance to efforts in managing heat. Thermal diodes have numerous applications, including blocking unwanted back scattering in phonon-based information processing as well as managing heat and maximizing efficiency in nanostructures. The operation of a thermal diode requires a nonlinear material or broken time-reversal symmetry[1]; most implementations have exploited the former[2–6].

Theoretical and experimental advances in our understanding of the contribution of coherent phonons to heat transport has provided new insights that allow for enhanced control of heat flow in nanostructured materials[7]. Specifically, due to the very long mean free paths and coherence of these phonons[8,9], periodic structures can modify their dispersion and transport properties[10]. Thus, engineered bandgaps, which have been used to manipulate sound[11], can also be used to alter the thermal properties of a material[12]. Moreover, adding impurities have been proven useful in modifying thermal transport properties of a material by manipulating high-energy phonons[13,14].

At the same time, there have been remarkable advances in cavity optomechanics[15], where interactions between photons and acoustic phonons confined in an optomechanical cavity can be controlled at the single phonon level[16], with potential applications in quantum information processing[17,18]. More recently, non-reciprocal optical transport was proposed in ring resonators, where the directional laser pump selects one circulation direction[19]. This scheme and approaches based on stimulated Brillouin scattering in photonics[20,21] were experimentally demonstrated in multiple optomechanical systems[22–26]. Meanwhile, the resulting chirality for phonons in such ring resonators has been investigated[27–29]. Moreover, there have been intriguing proposals to synthesize gauge fields in optomechanical systems, from photonic crystals[30,31], to quantum wells[32], and superconducting circuits[33], and to investigate their associated topological properties[34].

In this article, we combine the physics of heat transport in nanostructures and optomechanics to develop a new platform to manipulate both low-energy and high-energy phonons. We propose a method to engineer a tunable non-reciprocal bandgap for acoustic phonons, where a laser field with a phase gradient optically drives an array of optomechanical cavities and induces the non-reciprocal transport by breaking the time-reversal symmetry. We propose an experimental implementation of the scheme in optomechanical crystals. We discuss two applications of such a system, one as an acoustic isolator, and the other as a thermal diode. For the latter, the introduction of alloy and nanoparticle disorder suppresses the transport of high-energy phonons, leaving the low-energy acoustic band as the dominant channel for heat conduction[10,35], thus enhancing the overall optomechanically induced non-reciprocity. Our proposed device works in the linear regime and introduces an alternative approach to previous works.

## Results

**Tight-binding model**. To illustrate the basic concepts, we study non-reciprocal transport in a system connected to heat baths. The system considered here is described by a tight-binding model of an array of coupled optomechanical cavities[36–38]. An opto-mechanical cavity supports localized electromagnetic and mechanical modes. Owing to the radiation pressure, changes to the shape of the cavity change its electromagnetic resonance

frequency, effectively coupling mechanical vibrations to electromagnetic excitations. The Hamiltonian describing a collection of isolated cavities (setting $\hbar = 1$) is

$$\hat{H}_{\text{sites}} = \sum_n \omega_{\text{cav}} \hat{a}_n^\dagger \hat{a}_n + \omega_{\text{mech}} \hat{b}_n^\dagger \hat{b}_n - g\hat{a}_n^\dagger \hat{a}_n (\hat{b}_n^\dagger + \hat{b}_n), \qquad (1)$$

where $\hat{a}_n(\hat{b}_n)$ is a bosonic operator that destroys a photonic (phononic) excitation with energy $\omega_{\text{cav(mech)}}$ at site $n$, and $g$ is the vacuum coupling rate. In addition, there is a loss rate $\gamma_{\text{cav}}(\gamma_{\text{mech}})$ associated with the optical (mechanical) mode of the cavity (see Methods).

In a linear array of cavities, nearest-neighbor couplings dominates due to the tunneling of excitations between adjacent sites. The Hamiltonian describing these processes is

$$\hat{H}_{\text{tunneling}} = -J \sum_n \hat{a}_n^\dagger \hat{a}_{n+1} - t \sum_n \hat{b}_n^\dagger \hat{b}_{n+1} + \text{h.c.}, \qquad (2)$$

where h.c. denotes Hermitian conjugate, and $t$ and $J$ are tunneling strengths of phononic and photonic excitations, respectively. The system Hamiltonian is then given by

$$\hat{H}_{\text{sys}} = \hat{H}_{\text{sites}} + \hat{H}_{\text{tunneling}}. \qquad (3)$$

The vacuum coupling rate, $g$, is typically small, and can be enhanced by means of an external laser drive. To break reciprocity in a spatially dependent manner, in contrast to the usual setup in which the optical mode on each site is excited by a laser with a uniform phase[15], we consider a phase gradient in the laser field. This phase, which breaks time-reversal symmetry introduces a position dependent phase in the effective coupling between phonons and photons[31]. The breaking of time-reversal symmetry is crucial for the effects we consider in this work. The laser frequency, $\omega_{\text{d}} = \omega_{\text{cav}} + \Delta$, is detuned from the resonance frequency of the cavity by $\Delta$, and the phase offset between adjacent sites is $\theta$, as shown in Fig. 1b.

To bring electromagnetic and mechanical excitations on resonance, the driving laser is red-detuned from the cavity resonance frequency by $\Delta \approx -\omega_{\text{mech}}$. In a rotating frame of photons with angular frequency $\omega_{\text{d}}$, after making the rotating-wave approximation (RWA) in the resolved sideband regime ($\omega_{\text{mech}} \gg \gamma_{\text{mech}}$), linearizing and displacing the cavity field, the effective Hamiltonian is[39]

$$\begin{aligned} \hat{H}_{\text{eff}} = &-\Delta/2 \sum_n \hat{a}_n^\dagger \hat{a}_n + \omega_{\text{mech}}/2 \sum_n \hat{b}_n^\dagger \hat{b}_n - J \sum_n \hat{a}_{n+1}^\dagger \hat{a}_n \\ &- t \sum_n \hat{b}_{n+1}^\dagger \hat{b}_n - G \sum_n e^{-in\theta} \hat{a}_n^\dagger \hat{b}_n + \text{h.c.}, \end{aligned} \qquad (4)$$

where $G = \alpha g$, the enhanced optomechanical coupling strength is large compared to $g$ by an order of $\alpha$, the square root of the number of photons in the cavity.

The propagating modes of the system are polaritons, which are superpositions of electromagnetic and mechanical quanta. To diagonalize the Hamiltonian in Eq. (4), we write the Fourier transform for $\hat{a}_n$ and $\hat{b}_n$ as

$$\begin{pmatrix} \hat{a}_n \\ \hat{b}_n \end{pmatrix} = \sum_k e^{-iknd_0} \begin{pmatrix} \hat{a}_k \\ \hat{b}_k \end{pmatrix}, \qquad (5)$$

where $d_0$ is the lattice constant. Then Eq. (4) in the Fourier basis is

$$\hat{H}_k = \begin{pmatrix} -\Delta - 2J\cos(d_0 k - \theta) & -G \\ -G & \omega_{\text{mech}} - 2t\cos(d_0 k) \end{pmatrix}, \qquad (6)$$

which shows the phase gradient of the laser field acts as a

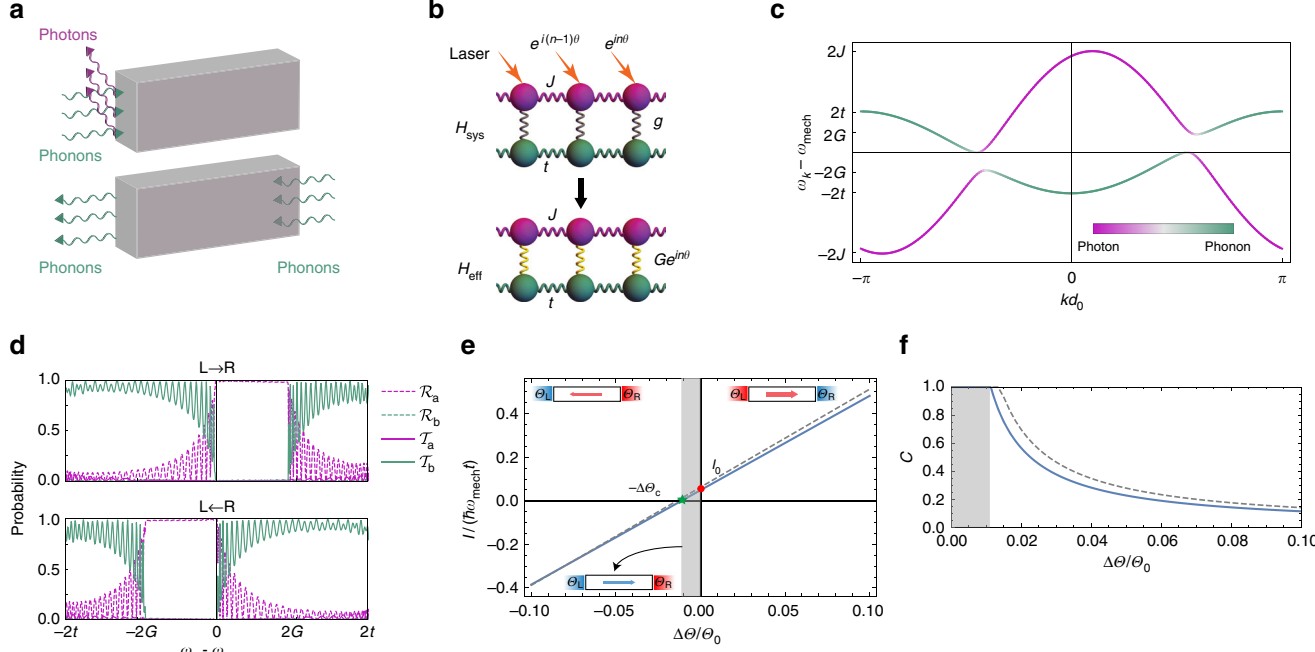

**Fig. 1** Sketch of the system and its transport properties. **a** The non-reciprocal device allows transmission of phonons in one direction, and converts them into photons, which are reflected in the opposite direction. **b** Schematic representation of the system, showing the coupling of phonon (green) and photon (pink) degrees of freedom and their hopping strengths $t$ and $J$, respectively. Adding a driving laser with a phase $e^{in\theta}$ to the bare Hamiltonian $H_{sys}$ (3) with optomechanical coupling $g$ leads to the effective Hamiltonian $H_{eff}$ (4) with an enhanced and position dependent optomechanical coupling $Ge^{in\theta}$. **c** The band structure corresponding to the Hamiltonian in eq. (6) for parameters $2G/t = 1$, $2J/t = 5$, and $\theta = 1.1\pi$ ($k$ is the wavenumber appearing in the eigenmodes $\alpha_{k,j}\hat{a}_{k-\theta} + \beta_{k,j}\hat{b}_k$). The color scale indicates the extent of phonon (green) or photon (pink) character of the eigenstate. **d** Transmission $\mathcal{T}_{a(b)}$ and reflection $\mathcal{R}_{a(b)}$ probabilities of photons (phonons) for right-moving (L → R) and left-moving (L ← R) phonons through a system with $N = 100$ sites plotted as a function of incident energy. The gaps in **c**, determine the energy range for which phonons are reflected from the system. The mismatch in these energy ranges for left- and right-moving phonons is the origin of the non-reciprocal transport. **e** The current $I$ as a function of temperature bias for the same parameters as **c**, and $k_B\Theta_0/\hbar\omega_{mech} = 1.5$. The non-reciprocity is evident in the non-zero intercept of the line in $I$-$\Delta\Theta$ plot. A key feature is that a non-zero current $I_0$ flows even in the case of zero bias. When the bias is $-\Delta\theta_c$ the current is extinguished. **f** Contrast $C$ as a function temperature bias $\Delta\Theta$. The shaded region in **e**, **f** corresponds to the case with $C = 1$, in which if the bias is reversed, the direction of the current is unchanged. The solid lines in **e**, **f** correspond to the Eqs. (8) and (12), whereas the dashed lines represent approximate expressions (13) and (14)

momentum shift, and leads to the coupling of phonons and photons with different momenta.

The band structure is shown in Fig. 1c. In the absence of coupling the phononic and photonic dispersions intersect. For $G \neq 0$, two bandgaps develop. The asymmetry (under $k \to -k$) of the band structure is controlled by $\theta$. The eigenmodes are polaritons, $\alpha_{k,j}\hat{a}_{k-\theta} + \beta_{k,j}\hat{b}_k$, where $j \in \{1, 2\}$ is the band index. The quantities $|\alpha_{k,j}|^2$ and $|\beta_{k,j}|^2$ indicate the relative weights of photons and phonons composing the polaritons, respectively. The effect of cavity loss is to broaden the bands, and for the gaps to be effective, we need to be in the high cooperativity regime, i.e., $G^2/\gamma_{cav}\gamma_{mech} \gg 1$.

The asymmetry of the gaps controls the non-reciprocal transport properties of the system, as shown Fig. 1d. To study thermal transport properties of this model, we consider connecting the system to two thermal contacts. The contacts are impedance matched to the non-driven ($G = 0$) system. Thus, the dispersion of phonons in the contacts is $\omega_{contact}(k) = \omega_{mech} - 2t\cos(d_0 k)$. The transmission probabilities can be calculated by mode matching at the boundaries of the system; see Fig. 1d and the Supplementary Note 1. This continuum picture remains valid for a finite number of lattice sites and the transmission probabilities are close to zero for phonons with energies in the gap. These phonons are converted to photons and reflected. The probabilities exhibit Fabry–Perot oscillations whose period is proportional to the inverse of the number of sites in the system.

The direction dependent phonon transmission probability for $\theta$ close to $\pi$ can be approximated by

$$\mathcal{T}_{L \rightleftarrows R}(\omega) = |H(\omega - \omega_{mech}) - H(\omega - \omega_{mech} \pm 2G)|, \quad (7)$$

where $H(\omega)$ is the Heaviside step function.

The phonon thermal current, assuming that the photon contacts are at zero temperature, can be calculated in the Landauer–Büttiker formalism[40,41], and is given by

$$I(\Theta_L, \Theta_R) = \int_0^\infty \frac{d\omega}{2\pi} \hbar\omega [\mathcal{T}_{L\to R}(\omega) n_B(\Theta_L, \omega) \\ - \mathcal{T}_{L\leftarrow R}(\omega) n_B(\Theta_R, \omega)], \quad (8)$$

where $\Theta_{L(R)}$ is the temperature of the left(right) contact, and $n_B(\Theta, \omega) = 1/(\exp(\hbar\omega/k_B\Theta) - 1)$. We introduce an alternative set of variables to denote the mean temperature, $\Theta_0$, and the temperature bias, $\Delta\Theta$, such that

$$I(\Theta_L, \Theta_R) = I(\Theta_0 + \Delta\Theta/2, \Theta_0 - \Delta\Theta/2), \quad (9)$$

**a**

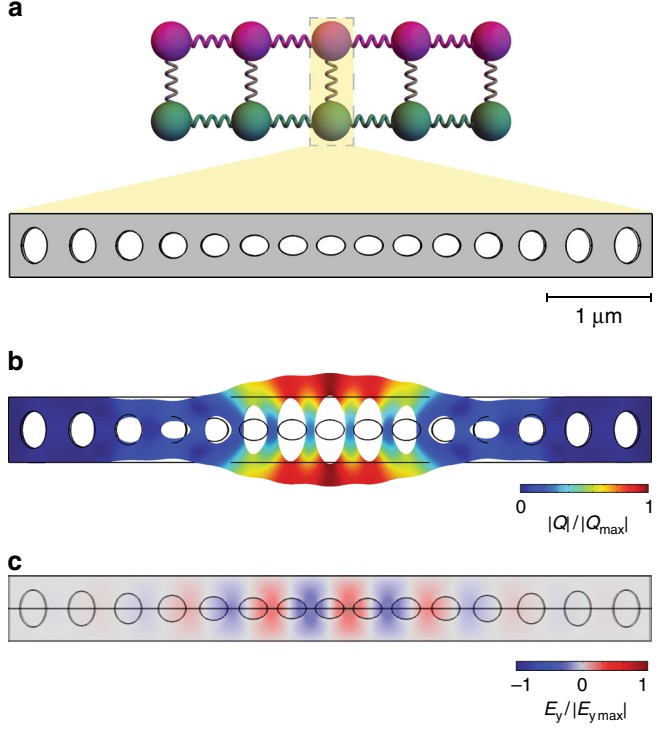

1 μm

**b**

0 $|Q|/|Q_{max}|$ 1

**c**

−1 0 1

$E_y/|E_{y_{max}}|$

**Fig. 2** The portion of an optomechanical crystal corresponding to a single site in the tight-binding model. **a** The correspondence between the ball and the physical realization (expanded view). **b** The same portion of the optomechanical crystal showing the normalized mechanical displacement ($|Q|/|Q_{max}|$) of a confined eigenmode, and **c**, the normalized electric field ($E_y/|E_{y_{max}}|$) of an eigenmode. The frequencies and coupling strengths can be calculated using finite-element (FEM) simulations (see Supplementary Note 2)

with

$$\Delta\Theta = \Theta_L - \Theta_R, \tag{10}$$

$$\Theta_0 = \frac{\Theta_L + \Theta_R}{2}. \tag{11}$$

This relationship implies that if $\Theta_0$ is fixed and only the sign of $\Delta\Theta$ is changed, the temperatures of the two contacts are swapped. In a reciprocal system, there is no distinction between left and right, and taking $\Delta\Theta \to -\Delta\Theta$ only changes the sign of the current and leaves the magnitude unchanged. However, due to the broken time-reversal symmetry in our system, transport is non-reciprocal $\mathcal{T}_{L\to R}(\omega) \neq \mathcal{T}_{L\leftarrow R}(\omega)$, and the current magnitudes are different. Figure 1e shows the current $I$ as a function of temperature bias $\Delta\Theta$. The base temperature $\Theta_0$ is chosen close to the energy scale $\omega_{mech}$ of the system, so that the non-reciprocal effect is enhanced. In this plot, the non-zero intercept ($I_0 \propto G^2$) is a measure of the non-reciprocity. This is different from an electrical diode mechanism, where the slope changes as the bias is reversed.

To quantify the non-reciprocity, we introduce the contrast $C$, defined as

$$C(\Theta_1, \Theta_2) = \frac{|I(\Theta_1, \Theta_2) + I(\Theta_2, \Theta_1)|}{|I(\Theta_1, \Theta_2)| + |I(\Theta_2, \Theta_1)|}, \tag{12}$$

which is non-zero in a non-reciprocal system. In the shaded region in Fig. 1e and f, the current does not change its direction

when the bias is reversed. In this case, the contrast is maximized and $C = 1$.

The relation between $I$ and $\Delta\Theta$ for $\hbar\omega_{mech}/k_B\Theta_0 \ll 1$, and $\Delta\Theta \ll \Theta_0$ is well described by

$$I(\Theta_L, \Theta_R) \approx 2k_B(2t - G)\Delta\Theta + 2\hbar G^2. \tag{13}$$

In the same regime, the contrast is given by

$$C \approx \begin{cases} 1 & \text{if } \Delta\Theta \leq \Delta\Theta_c \\ \hbar G^2/[k_B(2t - G)\Delta\Theta] & \text{otherwise} \end{cases}, \tag{14}$$

where $\Delta\Theta_c = \hbar G^2/[k_B(2t - G)]$. These approximations are compared with the exact values in Fig. 1e and f.

This non-reciprocal model can be implemented in an optomechanical crystal[42,43]. An optomechanical crystal is an engineered dielectric, which supports localized phononic and photonic excitations with energies in the bandgaps. Given a uniform dielectric, bandgaps can be introduced by drilling a periodic array of identical holes. Deforming these holes to form a superlattice introduces defect cavities which co-localize phonons and photons, thereby enhancing their mutual couplings. In Fig. 2, we show the correspondence between a cavity and its implementation in the actual optomechanical crystal. Each unit cell is a few microns in size, and a total system size of hundreds of microns leads to the non-reciprocity shown in Fig. 1d. The bare optomechanical coupling $g$ varies between several kilohertz and tens of megahertz in various materials such as Si or GaAs[26,44,45]. The tunneling strengths depend on the structure design, and values of a few megahertz for the mechanical tunneling strength $t$, and hundreds of megahertz for its optical counterpart, have been realized in the experiments[26]. In Methods and Supplementary Note 2, we present more details, and specifically show how to engineer the non-reciprocal band in an optomechanical crystal. The tight-binding model is applicable not only to optomechanical crystal arrays, but also to other optomechanical systems such as coupled ring resonators that have been realized in experiments[46].

**Applications**. Now that we have established that non-reciprocal transport for a continuum band of phonons can be achieved in an array of optomechanical cavities, we further discuss two applications in an optomechanical crystal (see Methods): (1) An acoustic isolator, and (2) a thermal diode in low temperatures.

An acoustic isolator is a device that only permits propagation of coherent monochromatic phonons in one direction. Such a device can be realized using a nonlinear medium attached to a phononic crystal[47,48], or through spatio-temporal modulation of system properties in a transmission line[49]. Our optomechanical crystal operates as an isolator for frequencies which lie in the bandgap. An appropriate figure of merit analogous to the contrast in Eq. (12) for monochromatic waves is

$$C_{iso}(\omega) = \left| \frac{\mathcal{T}_{L\to R}(\omega) - \mathcal{T}_{L\leftarrow R}(\omega)}{\mathcal{T}_{L\to R}(\omega) + \mathcal{T}_{L\leftarrow R}(\omega)} \right|, \tag{15}$$

The device discussed in the previous section, with the same red-detuned laser drive acts as an isolator for phonons with frequencies close to $\omega_{mech}$. Specifically, in the high cooperativity regime, the effect of loss is negligible, and we can use the transmission probabilities shown in Fig. 1d. We see that for propagating elastic waves with frequencies inside the bandgap, $C_{iso}(\omega)$ approaches unity, and otherwise, is very close to zero, thus realizing an isolator with a bandwidth of 2G. The non-reciprocity in our scheme is tunable, and the frequency range of the gap is also controllable and depends on the phase gradient of the laser

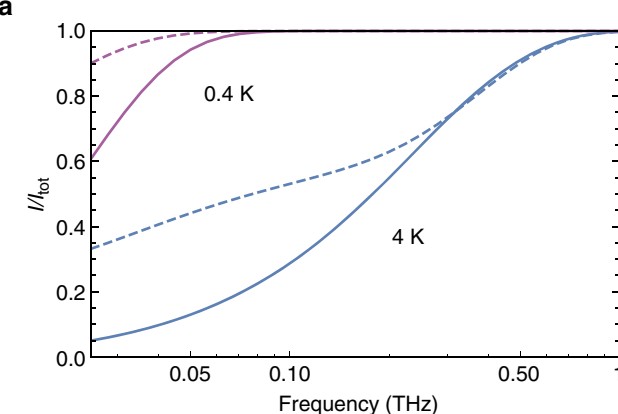

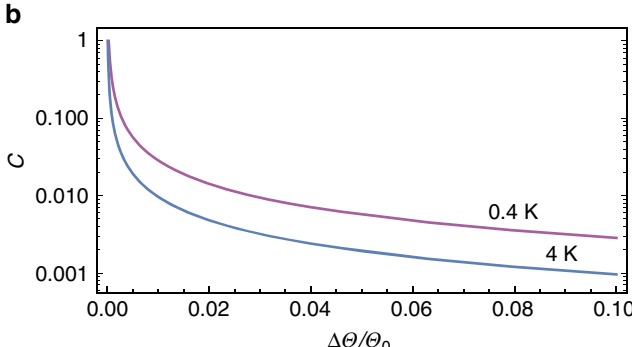

**Fig. 3** Thermal current and contrast. **a** Cumulative current ($I/I_{tot}$) in a beam of nonporous silicon (solid), compared with $Si_{90}Ge_{10}$ with nanoparticles optomechanical crystal (dashed), as a function of phonon frequency at $\Theta_0$ = 4 K (blue) and $\Theta_0$ = 0.4 K (purple) for a small temperature bias $\Delta\Theta/\Theta_0$ = $10^{-3}$. **b** Contrast $C$, defined in Eq. (12), as a function temperature bias $\Delta\Theta$ for an optomechanical cavity array made of $Si_{90}Ge_{10}$ with 10 nm nanoparticles at $\Theta_0$ = 0.4 K (purple), and 4 K (blue). We observe that the contrast increases as the temperature is decreased, because the optomechanically coupled phonons have a more pronounced role in the thermal transport at lower temperatures

field. Furthermore, as the Hamiltonian in Eq. (4) is linear in both the optical and mechanical fields, in principle the device works at the quantum limit. It is therefore useful for quantum information routing[18], and may find new applications to hybrid devices such as superconducting qubits[50] coupled to optomechanical crystals[17], as they both work in the same energy regime.

As a second application, we show that our system can serve as a thermal diode. A perfect thermal diode would allow heat transport in only one direction. Our system relies on the modification of the material properties in a narrow frequency range at low energies, whereas a major part of heat current is carried out by high-frequency phonons. To suppress the contribution of these high-energy phonons, we introduce various scattering mechanisms to shorten their mean free paths[10].

Specifically, to evaluate the figure of merit $C$ in a realistic material, we analyze frequency-dependent phonon scattering processes characterized by a length scale $\lambda_{ph}(\omega)$. The total transmission at a given frequency, summed over all bands, can be approximated as

$$\mathcal{T}_{L \rightleftarrows R}(\omega) = \frac{\lambda_{ph}(\omega)}{\lambda_{ph}(\omega) + L_s} f(\phi) M_{L \rightleftarrows R}(\omega), \qquad (16)$$

where the factor $\frac{\lambda_{ph}}{\lambda_{ph} + L_s}$ is the probability of transmittance[51,52], and $M_{L \rightleftarrows R}(\omega)$ is the number of conducting bands at a given energy for

right (L → R), or left (L ← R) moving phonons, and $\phi$ is the sample's porosity. The function $f(\phi) = \frac{1 - \phi}{1 + \phi}$ comes from Maxwell–Garnett effective medium approach, and takes the effect of holes on the number of modes into account[53,54]. The parameter $L_s$ is the length of the sample, and $\lambda_{ph}$ is the mean free path of back scattering, which is related to the mean free path of scattering, $l_{ph}$, by $\lambda_{ph} = 2l_{ph}$ in 1D, and $\lambda_{ph} = 4/3l_{ph}$ in 3D. Note that because of the dependence of $\lambda_{ph}$ on frequency, the performance of the device gets better for larger samples. Specifically, for larger samples the overall transmission decreases, however, as the mean free path is smaller for higher frequencies, the contrast improves. In our calculations, we considered $N$ = 100 sites.

Following refs. [10,55], we propose using alloy and nano-paricle impurities to modify $\lambda_{ph}$. In this case, the optomechanical crystal is made of an alloy into which nanoparticles are embedded. These impurities lead to mass-difference scatterings, and their respective rates $\tau_{alloy}^{-1}$ and $\tau_{np}^{-1}$ scales with $\omega^4$, thus lowering the contribution of high-energy phonons and increasing the contribution of low-energy phonons to thermal transport. To obtain a realistic estimate of $\lambda_{ph}$, it is necessary to consider two additional scattering mechanisms: intrinsic anharmonicity and boundary scattering, characterized by rates $\tau_{an}^{-1}$ and $\tau_b^{-1}$, respectively. The corresponding mean free paths $l_i$'s, are obtained from the scattering rates by $l_i = v_g \tau_i$, where $v_g$ is the group velocity. The total mean free path is given by Matthiessen's rule, i.e.,

$$\frac{1}{l_{ph}} = \frac{1}{l_{alloy}} + \frac{1}{l_{np}} + \frac{1}{l_b} + \frac{1}{l_{an}}. \qquad (17)$$

The effect of these scattering mechanisms on the cumulative thermal current, shown in Fig. 3a, is evaluated by a hybrid method using the bulk silicon dispersion ($\omega \propto k$) for high-energy phonons with short mean free paths and the superlattice dispersion for lower energy phonons with mean free paths longer than several lattice constants of the superlattice[54] (see Methods). It can be seen that phonons with frequencies below 25 GHz contribute more to heat transport in a $Si_{90}Ge_{10}$ optomechanical crystal with 10 nm nanoparticles and a filling factor of 5% than to a nanobeam of the same dimensions composed of nonporous silicon. Moreover, the ratio of the current carried by the optomechanically coupled band to the total current in the engineered optomechanical crystal is close to 9% at 4 K and about 22% at 0.4 K with a bias of $\Delta\Theta/\Theta_0 = 10^{-3}$ (see Supplementary Note 3).

Finally, we calculate the contrast for a driven optomechanical crystal, as displayed in Fig. 3b. As the base temperature decreases and approaches the energy of the optomechanical band ($k_B\Theta_0 \approx \hbar\omega_{mech}$), the contrast and the non-reciprocal effect increase. Although a significant constrast can be achieved at sub-Kelvin temperatures, the generalization of this scheme to room temperature requires a significant improvement in the material properties such as the optomechanical coupling strength (see Supplementary Note 3 for room temperature). Although there is intrinsic photon loss in the silicon beam, such loss does not directly lead to the generation of phonons in the beam[56], and therefore, the implicit assumptions of our approach remain valid (see Methods and Supplementary Note 3).

To measure the contrast, we envision a setup similar to ref. [57], where a pair superconductor/insulator/normal metal/insulator/ superconductor (SINIS) tunnel junction[58] are mounted at the two ends of the device and serve as both a sensitive thermometer and a heater. The device is heated from one side and the change in temperature is measured at the other side. By interchanging the role of the heater and thermometer and comparing the

measurements, the existence of non-reciprocal thermal current in the system can be verified.

## Discussion

In this work, we have shown that phase-modulated driven optomechanical systems can be utilized to engineer a non-reciprocal phonon band in a material. We discussed two possible applications of our scheme, as an acoustic isolator and a thermal diode. Although we considered a specific silicon-based optomechanical crystal, these methods can be readily generalized to other materials and designs. These devices may find application to on-chip heat management and quantum information processing, both to increase coherence time and to exploit phonons as information carriers[59].

In our approach, we proposed a coherent dynamical control of low-energy phonons by using optomechanical structures, and combined it with incoherent control of high-energy phonons by designing bulk material properties through the introduction of disorder. This strategy of combing low-energy and high-energy phononic physics could be generalized to designing of other thermal technologies such as thermoelectrics, thermal insulation, and the development of new metamaterials.

## Methods

**Implementation with optomechanical crystals**. In an optomechanical cavity electromagnetic and mechanical modes are co-localized. The coupling between these modes arises due to radiation pressure, which changes the cavity's electromagnetic resonance frequency. More rigorously, the energy $\hbar\omega(\hat{x})$ of a cavity depends on its shape, where $\hat{x} = x_{\mathrm{ZPF}}(\hat{b} + \hat{b}^\dagger)$ is the quantized mechanical displacement, and $x_{\mathrm{ZPF}}$ denotes zero point fluctuations. The Hamiltonian describing a single defect cavity is

$$H = \hbar\omega(\hat{x})\hat{a}^\dagger \hat{a} + \hbar\omega_{\mathrm{mech}}\hat{b}^\dagger \hat{b}. \tag{18}$$

The optomechanical coupling arises as the shape of the dielectric boundary changes. Expanding $\omega(\hat{x})$ to first order in $\hat{x}$ results in

$$\hat{H} = \hbar\omega_{\mathrm{cav}}\hat{a}^\dagger \hat{a} + \hbar\omega_{\mathrm{mech}}\hat{b}^\dagger \hat{b} - \hbar g \hat{a}^\dagger \hat{a}(\hat{b}^\dagger + \hat{b}), \tag{19}$$

where $g$ can be calculated from moving boundaries perturbation theory for Maxwell's equations[60,61]. Equation (19) reproduces the form of $\hat{H}_{\mathrm{sites}}$ (1).

Imperfect localization within each cavity leads to nearest-neighbor hopping of the phonons and photons as captured by $H_{\mathrm{tunneling}}$ (4). Using finite-element simulations (FEM), we find that to a very good approximation these bands follow the dispersion $\omega \propto \cos(k)$ as predicted by the tight-binding model (see Supplementary Note 2). Typical values of $\omega_{\mathrm{cav}}$, and $\omega_{\mathrm{mech}}$ for a silicon optomechanical crystal are 100 THz, and 10 GHz, respectively[26]. The laser frequency $\omega_{\mathrm{d}}$ should be $\omega_{\mathrm{cav}} - \omega_{\mathrm{mech}} \approx 100$ THz to bring it in resonance with the mechanical mode.

To linearize the Hamiltonian $\hat{H}_{\mathrm{sites}} + \hat{H}_{\mathrm{tunneling}}$ with the laser drive $\sum_n \epsilon_n e^{i\theta n} \cos(\omega_{\mathrm{d}}t)\,(a_n^\dagger + a_n)$, we solve for the steady-steady of the cavity in the absence of the optomechanical coupling ($g = 0$)[15], and find that the steady-state is given by

$$\alpha = \frac{\epsilon_{\mathrm{d}}}{\Delta - i\gamma_{\mathrm{cav}}/2 + 2J\cos(\theta)}, \tag{20}$$

where $|\alpha|^2$ is the number of photons in the cavity. Consequently, displacing the cavity field by $\hat{a}_n \rightarrow \hat{a}_n + \alpha e^{i\theta n}$, and using RWA results in $\hat{H}_{\mathrm{eff}}$ (4).

The phases ($e^{in\theta}$) can be tuned off the chip by using stretchable fiber phase shifters[26]. An on-chip implementation is possible by using a $1 \times N$ multi-mode interferometer to divide the power, and meandered waveguides or zero-loss resonators to tune the phase (see Supplementary Note 4).

**Scattering rates**. The scattering rate associated with the alloy disorder of $Si_xGe_{1-x}$ is described by an effective mass-difference Rayleigh scattering[62,63]

$$\tau_{\mathrm{alloy}}^{-1} = x(1-x)A\omega^4, \tag{21}$$

where $A = 3.01 \times 10^{-41}$ s$^5$ [10,55] is a constant that depends on the alloy properties. Scattering due to nanoparticles can be described by interpolating between the long- and short-wavelength scattering regimes[55,64],

$$\tau_{\mathrm{np}}^{-1} = v_{\mathrm{g}}(\sigma_{\mathrm{s}}^{-1} + \sigma_{\mathrm{l}}^{-1})^{-1}\frac{f}{V}, \tag{22}$$

where $f$ and $V = 4\pi r^3/3$ are the filling fraction and the volume of nanoparticles, $v_{\mathrm{g}}$ is the magnitude of the group velocity of phonons, and

$$\sigma_{\mathrm{s}} = 2\pi r^2, \tag{23}$$

$$\sigma_{\mathrm{l}} = \pi r^2 \frac{4}{9}(\Delta D/D)^2(\omega r/v_{\mathrm{g}})^4. \tag{24}$$

Here, $r$ is the radius of nanoparticles, $\Delta D$ is the difference between particle and alloy densities, and $D$ is the alloy's density. In our calculations, we consider a filling fraction of $f = 0.05$, $\sigma_{\mathrm{s}} = 6.28 \times 10^{-16}$ m$^2$, and $\sigma_{\mathrm{l}} = 2.2 \times 10^{-48}$ m$^6 \times (\omega/v_{\mathrm{g}})^4$, corresponding to $r = 10$ nm germanium nanoparticles in $Si_{90}Ge_{10}$.

The anharmonic scattering rate, which takes both the normal and umklapp processes into account, is given by

$$\tau_{\mathrm{an}}^{-1} = BT\omega^2 e^{-C/T}, \tag{25}$$

where $B(T) = 3.28 \times 10^{-19}$ s K$^{-1}$ and $C = 140$ K for $Si_{90}Ge_{10}$[10,55]. Scattering from the boundaries of a thin film can be modeled by $l_{\mathrm{b}} = \frac{1+p}{1-p}t$[65,66], where $t$, the thickness of the sample, is the mean free path in the diffusive limit. The parameter $p$ is the probability that the scattering is specular. It takes the effect of surface roughness into account and depends on the phonon's wavelength. It is given by[65,67]

$$p = \exp\left(-\frac{16\pi^2\eta^2}{\Lambda^2}\right), \tag{26}$$

where $\Lambda$ is the wavelength of the phonons, and $\eta$ is the surface roughness of the sample, which is taken to be 1 nm in our calculations, and corresponds to an estimated surface roughness achieved in silicon thin film fabrications[51].

**Thermal current**. In Fig. 3, we compared the operation of our optomechanical crystal with a nanobeam of nonporous silicon. The thermal current in the latter can be calculated as follows. At the temperatures considered, only phonons with frequencies smaller than 3 THz are relevant for thermal transport. In this frequency range, only the acoustic branch contributes to thermal conductivity and the dispersion is approximately linear. Specifically, we employed the Debye dispersion, i.e., $\omega = c_{\mathrm{s}}|\mathbf{k}|$, where $c_{\mathrm{s}}$ is the sound velocity, and $\mathbf{k}$ is the wavevector. The density of modes is given by $M(\omega) = S3\omega^2/4\pi c_{\mathrm{s}}^2$, where $S$ is the cross sectional area. In addition, because the sample in this case is nonporous, $f(\phi) = 1$, and the only scattering mechanisms are scatterings due to surface roughness and crystal anharmonicities.

The thermal current for the proposed $Si_{90}Ge_{10}$ optomechanical crystal with nanoparticles is calculated using a hybrid method, depending on the frequency of phonons[54]. Specifically, The mean free path of phonons depends on their frequency. As the frequency of phonons is lowered, their mean free path becomes comparable with the superlattice spacing, and therefore, the bulk dispersion is no longer a good description. In our system this threshold frequency corresponds to 25 GHz. To get this number, using the silicon bulk dispersion and Eqs. (17) and (21–26), we find the frequency for which the mean free path is comparable to several lattice spacings.

For phonons with frequencies above the threshold, we use the bulk Debye dispersion and associated group velocities and density of states. For phonons with frequencies lower than this threshold, we use the superlattice dispersion, calculated by FEM simulations, which gives group velocities and density of states different from those of the bulk. The scattering rates are then calculated using the superlattice dispersion in this regime. Taken together, these account for the total reciprocal phonon contribution to the thermal current. Finally, we add the contribution of the single non-reciprocal optomechanically coupled band to the calculated current. For this single band, we have used $\theta = 1.3\pi$ and parameters $\omega_{\mathrm{mech}}/2\pi = 4.3$ GHz, $J/2\pi = 0.5$ GHz, $t/2\pi = 0.2$ GHz, and $G/2\pi = 0.1$ GHz, which leads to asymmetric gaps in 3.96 GHz $< \omega/2\pi <$ 4.13 GHz, and 4.47 GHz $< \omega/2\pi <$ 4.64 GHz for right-going, and left-going phonons, respectively.

**Data availability**. The data that support the findings of this study are available from the authors upon reasonable request.

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

## Acknowledgements

We thank Krishna Balram for providing the initial FEM simulations, Hirokazu Miyake for helping with the simulations, and Reza Ghodssi for providing access to computational resources. We also thank Sunil Mittal, Raphaël Van Laer, Amir Safavi-Naeini, and Oskar Painter for helpful discussions. The work was partially supported by Sloan Fellowship, YIP-ONR, the NSF PFC at the JQI.

## Author contributions

A.S. and W.D. formulated the transmission probabilities in the tight-binding model. A.S. performed the calculation and numerical simulations. K.E. and M.H. directed the study. All authors discussed and analyzed the results and contributed and commented on the manuscript.

## Additional information

**Competing interests:** The authors declare no competing interests.

