## [Peer Review File(PDF 259 kb) · Nature Communications]

Reviewers' comments:

Reviewer #1 (Remarks to the Author):

The authors present a tight-binding model to describe an optomechanical array system. Even though similar optomechanical arrays and continuum behavior have been examined previously (Ref. 1 below – perhaps it should be cited), this system is subject to an external laser drive with a phase gradient, which is shown to break time-reversal symmetry. The authors also propose two interesting applications, an acoustic isolator and a thermal diode. The combined analysis on the interplay between a non-reciprocal optomechanical system and heat transfer is of particular interest and I don't believe it has been analyzed before.

I find that while the writing is good overall, the narrative does have weaknesses at many points. e.g. in the introductory paragraph, “thermal diodes has...” and the last sentence of that paragraph could both be improved. More importantly, the technical issues that I list here are concerning/problematic enough so as to require a major revision prior to further consideration.

1. On page 1 right column the authors say that only low energy phonons are dominant for heat conduction in the presence of alloy and nanoparticle disorder, but there are no supporting references provided.

2. The authors do not establish a clear link between G , g , and θ in the narrative. In particular, it is surprising how θ appears in equation 4 without a clear pre- or post-definition. This needs to be cleared up properly in the manuscript even if it is available in the Supplement.

3. The structure proposed in Fig 2 is discretized, and yet the analysis performed throughout the paper is for continuum behavior. However, the distinctions that would arise in a discrete implementation of the continuum model are never discussed (e.g. discrete modes instead of bands). Thus, the Fig 2 structure remains improperly defended in the context of this work.

4. There are additional practical constraints with the design that the authors should discuss. (i) The key to the breaking of time-reversal symmetry, a laser drive with phase gradient, is missing from the example optomechanical crystal system. How would an experimentalist incorporate the required phase gradient into a laser drive in such a system, while controlling phase to good precision? (ii) The size of an acoustic/thermal diode based on the proposed optomechanical crystal design would be

gigantic (each crystal is several tens of microns typically), so to approach a near-continuum case with this structure seems rather impractical.

5. The authors claim that the system can be applied as a thermal diode, however, the contribution from the non-reciprocal part of the band structure is not likely to affect the total thermal conductivity or the heat current. In particular, if we use the numbers from the article, the isolation bandwidth is about 2G300MHz. Since all other phonon modes (0 to ~ THz) are not optomechanically coupled to the optical modes and only phonons in a relatively narrow frequency range exhibit such non-reciprocal behavior, the proposed thermal diode seems to be unrealistic. I suggest that the authors more clearly articulate (perhaps with a calculation) the contributions to the thermal conductivity and/or the thermal current from different phonon modes, specifically the non-reciprocal contribution versus the reciprocal contribution.

6. For obtaining a thermal diode, the authors suggest the use of Si_xGe_y alloys, potentially with added nanoparticles to first shut down the contribution of THz phonons to heat conduction. However, the actual structure is never properly discussed. Would it be the same optomechanical crystal? Where would the material be employed? How would the optical properties change with the use of this material?

7. On page 4 the authors say that their scheme “works at the quantum limit” but do not qualify this claim properly.

8. The literature review on nonreciprocal optomechanics in ring resonators (refs 24-26) does not correctly identify the earliest papers on the topic.

Minor comments:

1. Page 3, last line on the bottom left θ_0 should be Θ_0 .

2. Fig 2 caption says optomechanical (typo).

3. Si₉₀Ge₁₀ in Fig. 3 is labeled as Si₁₀Ge₉₀.

4. At the end of the Methods section on Thermal Conductivity, the authors cite numbers in GHz – I believe they actually meant rad/s since the numbers are multiplied by 2π .

5. I noticed that many of the cited papers are out of date (incorrect journal / arXiv availability) and are missing information (pages and dates).

[1] P. T. Rakich and F. Marquardt, "Quantum Theory of Continuum Optomechanics," arxiv:1610.03012.

Reviewer #2 (Remarks to the Author):

In the submitted manuscript titled "Thermal management and non-reciprocal control of phonon flow via optomechanics", the authors have presented an approach to engineer a phononic system using optomechanical coupling to create a non-reciprocal control over phononic flow. The authors state that the combination of approaches will allow for a of design an acoustic isolator and a thermal diode. While the idea presented is novel and interesting, in its current form especially with the limited focus on the applications of the presented idea I believe it would be of very limited interest to a broader audience of Nature Communications. The manuscript in its current form should be a great addition to a more physics specific journal.

There are other points that the authors should clarify.

1. The first part of the study develops a theoretical framework to create a non-reciprocal control over phononic flow. In the latter part, the authors use a specific material (Si90Ge10) to demonstrate the applicability. It would be useful for the reader to state the values of various parameters (e.g. G, J etc.) introduced in the first section for the selected materials.

2. Furthermore, it would be useful to understand the motivation of the authors behind selecting a certain material as the focus of their study.

3. The authors stated in Methods that for frequencies $<25\text{GHz}$, the correction factor from the Maxwell Garnett effective medium approach will be unity. The correction factor accounts for the 'material removal' effects on thermal transport. Why should this be any different for any phonon irrespective of its frequency?

4. The authors have attempted to engineer a heat spectra best suited for the presented design but details on the underlying assumptions are critically lacking. For instance, it is my understanding that the derivation of the band structure makes assumptions of a non-absorptive media (Fig. S4). How reasonable are such assumptions in the context of thermal applications with finite phonon-mean-free-paths? Is the Debye approximation and Matthiessens rule without considering surface specularly reasonable assumptions? Why/Why not? The values of relaxation times that have been used are not clear.

5. What are the operating temperature ranges for the acoustic isolator?

6. The difference between an acoustic isolator and thermal diode needs to be brought out clearly. Is it the difference between the frequency ranges where phonon transport is subsided? What frequency range modulations does the optomechanical coupling impact in both devices?

7. What is the justification for the use of the Debye approximation for frequencies below 3 THz in Silicon?

8. The choice of the frequency of 25 GHz to differentiate between coherent and quasiballistic regimes seems rather arbitrary. What is the dependence of the wavelength with the length scale needs to be highlighted clearly? What is the role of phonon mean free paths in differentiating the two regimes?

9. Bulk cumulative conductivity has been shown in Fig. 3(a). However this is not what is implemented in the thermal diode device (at the nanoscale). What is the cumulative conductivity contribution in the device when using Si₁₀Ge₉₀ with nanoparticles at the corresponding operational temperatures? Providing an analysis and contrasting it with other cases such as nanoporous silicon without optomechanical coupling would further strengthen the validity of the proposed model.

Reviewer #3 (Remarks to the Author):

This paper describes an optomechanical scheme to break time-reversal symmetry for low energy acoustic phonons, thereby enabling non-reciprocal phonon transport that could lead to acoustic isolators and thermal diodes. The authors report that substantial diodicities can be achieved at temperatures less than 4 K.

The paper presents an interesting result. I am not convinced it is of the level of advance required for Nature Communications for the following reasons:

-time-reversal symmetry breaking by temporal modulation is well-known for other systems like photons and also for acoustic vibrations (see e.g. 10.1038/srep09926). This paper is adding the idea that the time-modulation can be performed using optomechanical effects, certainly a nice idea but still an addition onto a concept that is already known.

-the bandwidth of the scheme is very small, and therefore it is hard to see how it will be relevant for thermal transport. As the authors' calculations show non-trivial diodicities can be achieved only at low temperatures and sufficiently large thermal contrasts where thermal phonon bandwidth is naturally limited by the occupation function and varies rapidly with temperature.

-The authors put a lot of emphasis onto the idea that nanoparticles, etc can restrict the bandwidth of phonons that carry heat. First of all, these ideas are not really "recent developments" as termed in the abstract, having been known for over 50 years. Second, even with such scattering processes the resulting bandwidth is not narrow enough to fit within the relevant band for non-reciprocal transport – exactly as shown in Fig 3a. I feel the discussion of nanoparticles, etc is not really relevant since in the end the thermal occupation function is really what is needed to freeze out as many high-energy phonons as possible.

-The paper would be more useful and powerful if the authors would give specific, detailed predictions for how an experimentalist could implement the scheme and verify the non-reciprocal transport. More emphasis on these considerations like dimensions of the structure, laser driving wavelengths (which is given on p6), etc would be nice.

We thank the reviewers for their constructive comments. We are glad to see that the referees find our approach of combining broken time-reversal symmetry with time modulation and heat transport novel and interesting. We should mention that most of the limitations that the referees have pointed out are related to material properties of existing systems, and there's no fundamental limit to prevents higher efficiencies to be achieved in the future. As can be seen in the literature, the strength of the optomechanical coupling and the quality factors of the cavities are constantly improving, and it's not unrealistic to assume that a proposal that currently illustrates proof of principle, would have major impact in the future.

Our detailed response to referees comments are as follows:

Reviewer #1:

The authors present a tight-binding model to describe an optomechanical array system. Even though similar optomechanical arrays and continuum behavior have been examined previously (Ref. 1 below – perhaps it should be cited), this system is subject to an external laser drive with a phase gradient, which is shown to break time-reversal symmetry. The authors also propose two interesting applications, an acoustic isolator and a thermal diode. The combined analysis on the interplay between a non-reciprocal optomechanical system and heat transfer is of particular interest and I don't believe it has been analyzed before.

I find that while the writing is good overall, the narrative does have weaknesses at many points. e.g. in the introductory paragraph, “thermal diodes has...” and the last sentence of that paragraph could both be improved. More importantly, the technical issues that I list here are concerning/problematic enough so as to require a major revision prior to further consideration.

We appreciate the referee's time in reading our manuscript and also for finding our work to be of “particular interest”. We have added the citation to the mentioned reference. Below, we respond to the specific points raised by the referee.

R1.1) On page 1 right column the authors say that only low energy phonons are dominant for heat conduction in the presence of alloy and nanoparticle disorder, but there are no supporting references provided.

A1.1) We thank the referee for pointing out the missing references, and added the corresponding citations.

R1.2) The authors do not establish a clear link between G , g , and θ in the narrative. In particular, it is surprising how θ appears in equation 4 without a clear pre- or post-definition. This needs to be cleared up properly in the manuscript even if it is available in the Supplement.

A1.2) The parameter g is defined between equations (3) and (4). Lower-case θ is defined before equation (3) and G is defined after equation (4) in the original manuscript. If the referee believes that we should further clarify these parameters we can revise the manuscript.

R1.3) The structure proposed in Fig 2 is discretized, and yet the analysis performed throughout the paper is for continuum behavior. However, the distinctions that would arise in a discrete implementation of the continuum model are never discussed (e.g. discrete modes instead of bands). Thus, the Fig 2 structure remains improperly defended in the context of this work.

A1.3) Initially we considered 15 sites/cells. We note that our conclusion is independent of the number of cells. However, in order to address referee's concern, we have now repeated the calculations for 100 sites in the revised manuscript. Specifically, figures 2, 3, and S3 are now modified in the revised manuscript. With 100 lattice sites, the spacing between the discrete points ($4t/100$) is a few MHz which is not distinguishable from the continuum in an experiment. We explicitly see that the results of the paper remains unchanged. Thanks to the referees comments, we have further clarified our choice of length, after equation (15).

R1.4) There are additional practical constraints with the design that the authors should discuss. (i) The key to the breaking of time-reversal symmetry, a laser drive with phase gradient, is missing from the example optomechanical crystal system. How would an experimentalist incorporate the required phase gradient into a laser drive in such a system, while controlling phase to good precision? (ii) The size of an acoustic/thermal diode based on the proposed optomechanical crystal design would be gigantic (each crystal is several tens of microns typically), so to approach a near-continuum case with this structure seems rather impractical.

A1.4) The phase shift can be implemented by using the same laser source for all the cavities and stretchable fiber phase shifters as implemented in Ref. 22. We thank the referee for asking for this clarification on the experimental realization and we have added a discussion in the main text. As mentioned in A1.3 all the calculations assumed 100 sites. Therefore, given that each unit cell is a few microns, the total system size would be hundreds of microns. We thank the referee for pointing out the missing discussion and added details about the system size in the main text.

R1.5) The authors claim that the system can be applied as a thermal diode, however, the contribution from the non-reciprocal part of the band structure is not likely to affect the total thermal conductivity or the heat current. In particular, if we use the numbers from the article, the isolation bandwidth is about 2G300MHz. Since all other phonon modes (0 to \sim THz) are not optomechanically coupled to the optical modes and only phonons in a relatively narrow frequency range exhibit such non-reciprocal behavior, the proposed thermal diode seems to be unrealistic. I suggest that the authors more clearly articulate(perhaps with a calculation) the contributions to the thermal conductivity and/or the thermal current from different phonon modes, specifically the non-reciprocal contribution versus the reciprocal

contribution.

A1.5) We agree with the referee that the bandwidth is restricted by G. However, this is a material property and it is likely to increase as the experimental technology improves. While the referee is correct in stating that typically at room temperature phonon modes are occupied between 0 to THz frequencies, we have shown that at lower temperatures, with our proposed controlled scattering mechanisms, the non-reciprocal band can play a more prominent role in thermal conductivity. Previously, we had a discussion of limitations at room temperature in the supplementary, but in the revised version we have added the discussion to the main text for further clarification. We have also calculated and added the contribution of non-reciprocal band versus the total thermal current as the referee suggested.

R1.6) For obtaining a thermal diode, the authors suggest the use of Si_xGe_y alloys, potentially with added nanoparticles to first shut down the contribution of THz phonons to heat conduction. However, the actual structure is never properly discussed. Would it be the same optomechanical crystal? Where would the material be employed? How would the optical properties change with the use of this material?

A1.6) Our proposed strategy is to make the optomechanical crystal out of SiGe alloy, and embed nano-particles in it. We have made it clear in the revised version. The optical properties of Si_xGe_{1-x} are well studied. Since we are considering a small ratio of germanium (x=0.9), the properties don't change significantly. In particular, the energy gap which determines absorption coefficient is decreased from 1.11 eV to around 1.05 eV, which is still above 0.8eV (corresponding to 1550nm) energy of the laser drive.

R1.7) On page 4 the authors say that their scheme “works at the quantum limit” but do not qualify this claim properly.

A1.7) Since the effective Hamiltonian in equation (4) is linear in both optical and mechanical field in principle the device works at the quantum limit. We thank the referee and added a sentence clarifying the claim.

R1.8) The literature review on nonreciprocal optomechanics in ring resonators (refs 24-26) does not correctly identify the earliest papers on the topic.

A1.8) To best of our knowledge, achieving non-reciprocity in micro-ring resonators using optomechanics was proposed in Ref. 19. If the referee has a suggestion, we would appreciate it.

Minor comments:

1. Page 3, last line on the bottom left θ_0 should be Θ_0 .
2. Fig 2 caption says optimechanical (typo).
3. Si₉₀Ge₁₀ in Fig. 3 is labeled as Si₁₀Ge₉₀.
4. At the end of the Methods section on Thermal Conductivity, the

authors cite numbers in GHz – I believe they actually meant rad/s since the numbers are multiplied by 2π .

5. I noticed that many of the cited papers are out of date (incorrect journal / arXiv availability) and are missing information (pages and dates).

We have addressed the minor comments mentioned by the referee.

Reviewer #2:

In the submitted manuscript titled “Thermal management and non-reciprocal control of phonon flow via optomechanics”, the authors have presented an approach to engineer a phononic system using optomechanical coupling to create a non-reciprocal control over phononic flow. The authors state that the combination of approaches will allow for a of design an acoustic isolator and a thermal diode. While the idea presented is novel and interesting, in its current form especially with the limited focus on the applications of the presented idea I believe it would be of very limited interest to a broader audience of Nature Communications. The manuscript in its current form should be a great addition to a more physics specific journal.

We appreciate that the referee finds our work interesting. Since our work draws upon two separate research fields, i.e. optomechanics and material science, we believe that our proposal is of interest to a wide audience of experimental and theoretical physicists and material scientists, interested in engineered non-reciprocal devices, from classical to quantum regime, and will provide a significant stimulus to this community.

R2.1) The first part of the study develops a theoretical framework to create a non-reciprocal control over phononic flow. In the latter part, the authors use a specific material (Si90Ge10) to demonstrate the applicability. It would be useful for the reader to state the values of various parameters (e.g. G, J etc.) introduced in the first section for the selected materials.

A2.1) We thank the referee for requesting clarification, and we have added a discussion in the in the last paragraph of the left column of page 4 in the revised manuscript and cited a range of relevant parameters in the text.

R2.2) Furthermore, it would be useful to understand the motivation of the authors behind selecting a certain material as the focus of their study.

A2.2) Our motivation for using silicon and SiGe is that they are CMOS compatible and silicon has been previously used in the fabrication of optomechanical crystals.

R2.3) The authors stated in Methods that for frequencies $<25\text{GHz}$, the correction factor from the Maxwell Garnett effective medium approach will be unity. The correction factor accounts

for the ‘material removal’ effects on thermal transport. Why should this be any different for any phonon irrespective of its frequency?

A2.3) We agree with the referee that the Maxwell-Garnet factor should not depend on the frequency. However, the reason that it is taken to be unity for frequencies less than 25 GHz in our work is that we are using a hybrid approach to calculate the thermal current (see ref. 49), and for those frequencies we have used the band structure of the superlattice. The calculations are done using a finite element method package COMSOL, and have taken the effect of holes into account. Therefore, there is no need to account for the porosity as it has already been accounted for in the band structure calculations.

R2.4) The authors have attempted to engineer a heat spectra best suited for the presented design but details on the underlying assumptions are critically lacking. For instance, it is my understanding that the derivation of the band structure makes assumptions of a non-absorptive media (Fig. S4). How reasonable are such assumptions in the context of thermal applications with finite phonon-mean-free-paths? Is the Debye approximation and Matthiessens rule without considering surface specularly reasonable assumptions? Why/Why not? The values of relaxation times that have been used are not clear.

A2.4) The referee makes an important point that a band structure calculation assumes no dissipation. We emphasize that the band structure of the superlattice is a starting point for our calculations, but in all energy regimes we take into account the finite lifetime of the phonons. Indeed, the superlattice band structure is used as input for the lifetime of phonons whose mean free path is larger than a few lattice spacing, The effect of surface specularly was accounted in our calculations by wavelength dependent l_b . We thank the referee his/her comments and we have completely revised the “Thermal current” section in the methods and emphasized these points. We also clarified the scattering rates in the corresponding section in the methods.

R2.5) What are the operating temperature ranges for the acoustic isolator?

A2.5) In order not to have thermal occupation of the optomechanically coupled phonon mode one has to work in the cryogenic regime.

R2.6) The difference between an acoustic isolator and thermal diode needs to be brought out clearly. Is it the difference between the frequency ranges where phonon transport is subsided?ke What frequency range modulations does the optomechanical coupling impact in both devices?

A2.6) This is an important point. Our system will act as both an acoustic isolator and a diode. An isolator only operates with monochromatic waves of a specific frequency. Our device will operate as an isolator for any frequency in the band-gap. In contrast, a thermal diode refers to the non-reciprocal transport behavior of a system with an applied temperature drop. We have added a clarification of this terminology in our revised manuscript.

R2.7) What is the justification for the use of the Debye approximation for frequencies below 3 THz in Silicon?

A2.7) Only acoustic phonons have frequencies less than 3 THz. Moreover, the phonon dispersion of silicon (see figure 3 in ref. 9) is to a good approximation linear in this frequency range. The Debye approximation is thus justified.

R2.8) The choice of the frequency of 25 GHz to differentiate between coherent and quasiballistic regimes seems rather arbitrary. What is the dependence of the wavelength with the length scale needs to be highlighted clearly? What is the role of phonon mean free paths in differentiating the two regimes?

A2.8) The choice of 25 GHz is not arbitrary. In fact, 25GHz is the energy scale at which the mean free path becomes comparable to superlattice spacing. Therefore, for phonon energies below this value, the bulk properties (Debye dispersion) can be used. However, for phonons above this value, the superlattice structure becomes important.

We agree with the referee that in our initial manuscript these points were not clear and were only discussed in the supplementary material. As discussed above in A2.4, we have completely revised the “Thermal current” section and clarified these points.

R2.9) Bulk cumulative conductivity has been shown in Fig. 3(a). However this is not what is implemented in the thermal diode device (at the nanoscale). What is the cumulative conductivity contribution in the device when using Si₁₀Ge₉₀ with nanoparticles at the corresponding operational temperatures? Providing an analysis and contrasting it with other cases such as nanoporous silicon without optomechanical coupling would further strengthen the validity of the proposed model.

A2.9) We thank the referee for the suggestion. We have replaced figure 3a with cumulative current at the corresponding temperatures to show the effect of alloy and nanoparticle impurities in an optomechanical crystal and compare the result with nonporous silicon. We can see that the use of alloy and nanoparticles increase the share of phonons with frequencies below 25 GHz from about 5% to 35% at 4K, and from about 60% to 90% at 0.4K.

Reviewer #3:

This paper describes an optomechanical scheme to break time-reversal symmetry for low energy acoustic phonons, thereby enabling non-reciprocal phonon transport that could lead to acoustic isolators and thermal diodes. The authors report that substantial diodicities can be achieved at temperatures less than 4 K.

The paper presents an interesting result. I am not convinced it is of the level of advance

required for Nature Communications for the following reasons:

We appreciate the referee's comments and in particular finding that our results are interesting. Below, we respond to the specific points raised by the referee.

R3.1) time-reversal symmetry breaking by temporal modulation is well-known for other systems like photons and also for acoustic vibrations (see e.g. 10.1038/srep09926). This paper is adding the idea that the time-modulation can be performed using optomechanical effects, certainly a nice idea but still an addition onto a concept that is already known.

A3.1) We appreciate the referee's comments regarding the previous works on time-reversal symmetry broken systems. Our proposal is indeed motivated by other time-breaking schemes which have been realized in various physical systems. We have cited the paper which is pointed out by the referee, along with other proposals and experimental realizations. As mentioned above, the goal of our work is to present a new paradigm to control non-reciprocal phonon transport, in the acoustic domain and beyond, by combining the field of heat management in nanostructured material and optomechanics. Specifically, we develop a method to engineer a tunable non-reciprocal band of acoustic phonons, by exploiting optomechanics. Moreover, building upon and expanding the toolbox for engineering phonons with higher energies and controlling thermal transport, we suppress the share of high-energy phonons in heat transport and control the heat flow by using optomechanics with low-energy phonons, while staying in the linear regime.

R3.2) the bandwidth of the scheme is very small, and therefore it is hard to see how it will be relevant for thermal transport. As the authors' calculations show non-trivial diodicities can be achieved only at low temperatures and sufficiently large thermal contrasts where thermal phonon bandwidth is naturally limited by the occupation function and varies rapidly with temperature.

A3.2) We agree that the contrast achieved in our proposal is limited by the bandwidth. However, we believe that there is a considerable need for non-reciprocal devices, even at low temperatures/narrow bandwidth. Moreover, there is no fundamental limitation to the size of G , and thus we anticipate that the operational bandwidth could increase as new materials become available. These advances would increase the operational range of temperature of our scheme.

R3.3) The authors put a lot of emphasis onto the idea that nanoparticles, etc can restrict the bandwidth of phonons that carry heat. First of all, these ideas are not really "recent developments" as termed in the abstract, having been known for over 50 years. Second, even with such scattering processes the resulting bandwidth is not narrow enough to fit within the relevant band for non-reciprocal transport – exactly as shown in Fig 3a. I feel the discussion of nanoparticles, etc is not really relevant since in the end the thermal occupation function is

really what is needed to freeze out as many high-energy phonons as possible.

A3.3) To address referee's comment, we have removed the word "recent". We note that although the effect of alloy and nano-particles on thermal conductivity has been known for many year, it is our understanding that only recently it was proposed to use them in phononic crystals for controlled engineering of heat flow (e.g. ref. 10). We disagree with the referee that nanoparticles are irrelevant. We have revised figure 3a to more clearly show these effects at the our operational temperatures. Indeed, as shown in figure 3a, the addition of nanoparticles has significant effect on thermal current and dramatically improves the performance of the device.

R3.4) The paper would be more useful and powerful if the authors would give specific, detailed predictions for how an experimentalist could implement the scheme and verify the non-reciprocal transport. More emphasis on these considerations like dimensions of the structure, laser driving wavelengths (which is given on p6), etc would be nice.

A3.4) We thank the referee for this suggestion. We have added more details about the design, dimension, and relevant parameters in the last paragraph of the left column of page 4 in the revised manuscript. We have also added a paragraph on how one can measure such a thermal current in our setup in the last paragraph of the right column of page 5.

Reviewers' comments:

Reviewer #1 (Remarks to the Author):

The authors have successfully addressed most of my comments, however a few items remain.

Regarding A1.4 (practicality) ---

I am still rather dissatisfied with the authors' proposal on how a suitable optical drive phase gradient could be obtained. A stretchable fiber phase shifter is a very large device (many centimeters) and cannot provide $N=100$ unique phases for a compact optomechanical system. It absolutely cannot work for the continuum case. Consider also the fact that typical fibers (i.e. the output of the fiber phase shifter) are 50-150 μm in diameter and the proposed device unit cell is single-digit micron scale.

In fact, I cannot even visualize a sufficiently compact photonic integrated circuit that could provide such a large number of distinct phases with a reasonable footprint. A better approach would be to somehow internally generate the required phases from a single external optical drive (perhaps slow/fast light), and I really would like to have seen such a proposal. Ideally along with a sketch in the supplement on how it could be integrated into the system.

Regarding A1.8 (literature review) ---

Ref 19 (Hafezi/Rabl) is indeed the first theoretical proposal for nonreciprocal cavity optomechanics. However the following two experimental cavity optomechanical papers appeared earlier than Refs 20-22, and just a few weeks apart.

J. Kim et al. Nat Phys 11, pp. 275, 2015.

C. Dong et al. Nat Comms doi:10.1038/ncomms7193, 2015.

The authors might also consider citing the following works, that while not concerning cavities, are still optomechanical, nonreciprocal, linear waveguide systems.

(theory) X. Huang and S. Fan, *J. Lightwave Technol.* 29, 2267–2275, 2011.

(expt) M.S. Kang et al, *Nat Photon* 5, 549–553, 2011.

I still think the work is good, but the above points do need to be addressed in a much better way.

Reviewer #2 (Remarks to the Author):

Dear Editor,

I am satisfied with most of the answers by the authors. Some minor comments are included below which can make the manuscript stronger if addressed by the authors.

In the “Thermal Current” section the authors seem to have made a typo/mistake as they state that “Specifically, for phonons with frequencies greater than 25 GHz, the associated mean free path is larger than the superlattice spacing...”. This line should be mean free path less than the superlattice spacing.

The authors state:

A2.8) The choice of 25 GHz is not arbitrary. In fact, 25GHz is the energy scale at which the mean free path becomes comparable to superlattice spacing. Therefore, for phonon energies below this value, the bulk properties (Debye dispersion) can be used. However, for phonons above this value, the superlattice structure becomes important. We agree with the referee that in our initial manuscript these points were not clear and were only discussed in the supplementary material. As discussed above in A2.4, we have completely revised the “Thermal current” section and clarified these points.

Following up, the authors have justified their choice of the frequency 25GHz but since their model is surface roughness dependent, this particular frequency will be influenced by the roughness. Some comment would help to generalize the model for various structural conditions.

Minor errors: Multiple typos in spellings in the manuscript.

In summary, the paper can be considered for publication.

Reviewer #3 (Remarks to the Author):

The authors have made a number of improvements to the paper. I now believe it is suitable for publication.

We thank the reviewers for their positive feedback. We are glad to see that the referees are mostly satisfied with the revised manuscript.

Our detailed response to referees' comments are as follows:

Reviewer #1:

The authors have successfully addressed most of my comments, however a few items remain.

We are glad that we have successfully addressed most of the referee's concerns.

R1.1) I am still rather dissatisfied with the authors' proposal on how a suitable optical drive phase gradient could be obtained. A stretchable fiber phase shifter is a very large device (many centimeters) and cannot provide $N=100$ unique phases for a compact optomechanical system. It absolutely cannot work for the continuum case. Consider also the fact that typical fibers (i.e. the output of the fiber phase shifter) are 50-150 μm in diameter and the proposed device unit cell is single-digit micron scale.

In fact, I cannot even visualize a sufficiently compact photonic integrated circuit that could provide such a large number of distinct phases with a reasonable footprint. A better approach would be to somehow internally generate the required phases from a single external optical drive (perhaps slow/fast light), and I really would like to have seen such a proposal. Ideally along with a sketch in the supplement on how it could be integrated into the system.

A1.2) We agree with the referee that fiber phase shifters are not suitable for an on-chip implementation. Indeed, we imagined a situation where these components are off the chip as in Fig3 of (Fang, Kejie, et al *Nature Physics* 13.5 (2017): 465). For N cavity system, $1 \times N$ splitters are available. Therefore, the same scheme can be generalized to our situation. However, we can think of another scheme where this phase can be integrated on-chip using $1 \times N$ multimode interferometers, for example what are used in arrayed waveguide gratings. In such a scenario, a phase gradient could easily be implemented using the propagation phase of the laser by varying the length of the waveguide, or by using zero-loss resonators as all-pass filters. In the latter, the phase depends on the resonance frequencies of the resonators and can be tuned by heating the resonators. We have expanded our discussion of the implementation of the phase in the right column of page 6 in the manuscript, and added a corresponding section in the supplementary section IV with a blue print of the on-chip approaches.

R1.2) Regarding A1.8 (literature review) ---

Ref 19 (Hafezi/Rabl) is indeed the first theoretical proposal for nonreciprocal _cavity_ optomechanics. However the following two experimental cavity optomechanical papers appeared earlier than Refs 20-22, and just a few weeks apart.

J. Kim et al. Nat Phys 11, pp. 275, 2015.

C. Dong et al. Nat Comms doi:10.1038/ncomms7193, 2015.

The authors might also consider citing the following works, that while not concerning cavities, are still optomechanical, nonreciprocal, linear waveguide systems.

(theory) X. Huang and S. Fan, J. Lightwave Technol. 29, 2267–2275, 2011.

(expt) M.S. Kang et al, Nat Photon 5, 549–553, 2011.

I still think the work is good, but the above points do need to be addressed in a much better way.

A1.2) We thank the referee for his/her suggestions, and cited the suggested references.

Reviewer #2

I am satisfied with most of the answers by the authors. Some minor comments are included below which can make the manuscript stronger if addressed by the authors.

We are happy that the referee is satisfied with the revised version of the manuscript.

R2.1) In the “Thermal Current” section the authors seem to have made a typo/mistake as they state that “Specifically, for phonons with frequencies greater than 25 GHz, the associated mean free path is larger than the superlattice spacing...”. This line should be mean free path less than the superlattice spacing.

A2.1) We thank the referee for spotting the typo and we have fixed it.

R2.2) The authors state:

A2.8) The choice of 25 GHz is not arbitrary. In fact, 25GHz is the energy scale at which the mean free path becomes comparable to superlattice spacing. Therefore, for phonon energies below this value, the bulk properties (Debye dispersion) can be used. However, for phonons above this value, the superlattice structure becomes important. We agree with the referee that in our initial manuscript these points were not clear and were only discussed in the supplementary material. As discussed above in A2.4, we have completely revised the “Thermal current” section and clarified these points.

Following up, the authors have justified their choice of the frequency 25GHz but since their model is surface roughness dependent, this particular frequency will be influenced by the roughness. Some comment would help to generalize the model for various structural conditions.

Minor errors: Multiple typos in spellings in the manuscript.

In summary, the paper can be considered for publication.

A2.2) The referee raises an important point. The threshold frequency depends on various structural and material properties. Indeed, when the boundary scattering is the limiting length

scale, it will set the relevant energies and the appropriate dispersion. In the hybrid method (see also Ref. 54 in the manuscript), we calculate the mean free path using silicon dispersion, and set the threshold frequency where this quantity is comparable to several lattice spacings. The intuition is that phonons can now sample the periodicity of the structure and 'see' the dispersion of the superlattice. We have added a discussion on the subject in the right column of page 7 in the revised manuscript.

Reviewer #3

The authors have made a number of improvements to the paper. I now believe it is suitable for publication.

We are glad that the referee sees our paper suitable for publication.